# scMAR-Seq: a novel workflow for targeted single-cell genomics of microorganisms using radioactive labeling

Hao-Yu Lo,[1,2] Konstantin Wink,[3] Henrike Nitz,[2] Matthias Kästner,[2] Detlev Belder,[3] Jochen A. Müller,[1,2] Anne-Kristin Kaster[1]

**ABSTRACT** Current methods for the identification of specific microorganisms based on an *in situ* metabolism are often hampered by insufficient sensitivity and habitat complexity. Here, we present a novel approach for identifying and sequencing single microbial cells metabolizing a specific organic compound with high sensitivity and without prior knowledge of the microbial community. The workflow consists of labeling individual cells with a $[^{14}C]$ substrate based on their metabolic activity, followed by encapsulating cells in alginate with nuclear emulsion by using microfluidics. We here adapted the concept of microautoradiography to visually distinguish between encapsulated labeled and non-labeled cells, which are then sorted via flow cytometry for single cell genomics. As a proof-of-concept, we labeled, separated, lysed, and sequenced single cells of the benzene degrader *Pseudomonas veronii* from mock microbial communities. The cells of *P. veronii* were isolated with 100% specificity. Single-cell microautoradiography and genome sequencing is an innovative method for elucidating microbial identity, activity, and function in diverse habitats, contributing to elucidate novel taxa and genes with potential for biotechnological applications such as bioremediation.

**IMPORTANCE** A central question in microbial ecology is which member of a community performs a particular metabolism. Several sophisticated isotope labeling techniques are available for analyzing the metabolic function of populations and individual cells in a community. However, these methods are generally either insufficiently sensitive or throughput-limited and thus have limited applicability for the study of complex environmental samples. Here, we present a novel approach that combines highly sensitive radioisotope tracking, microfluidics, high-throughput sorting, and single-cell genomics to simultaneously detect and identify individual microbial cells based solely on their *in situ* metabolic activity, without prior information on community structure.

**KEYWORDS** single-cell sequencing, microautoradiography, microbial metabolism, benzene degradation, FACS, microfluidics, metabolic activity

The metabolic activities of microorganisms are the basis for all life on Earth. They drive biogeochemical cycles, influence eukaryotic hosts, remediate contaminated sites, and play an important role in biotechnology (1–4). However, our current knowledge of microbial *in situ* activity is still mainly based on bulk analyses of microbiomes as well as deductions from cultivated isolates and laboratory enrichments that are considered representative (5). Since the vast majority of all microbial species have not yet been cultured (so-called microbial dark matter) (6–8), it is often uncertain how comprehensive and accurate our understanding of *in situ* functionality of a complex microbial community actually is, including which members catalyze a reaction of interest.

To close this knowledge gap, a suite of powerful labeling methods has been developed in which metabolically active microorganisms are traced by their incorporation of a stable or radioactive isotope and taxonomically identified (9–11). Meta-omics

Address correspondence to Anne-Kristin Kaster, kaster@kit.edu, or Jochen A. Müller, jochen.mueller@kit.edu.

The authors declare no conflict of interest.

approaches combined with stable isotope probing (SIP) have yielded a wealth of results in microbial ecology. Unfortunately, community-level SIP has comparatively low sensitivity when analyzing nucleic acids or protein sequences, or low taxonomic resolution in the case of lipids. In $^{13}$C-SIP, for example, at least 2%–10% atom incorporation is needed for the isopycnic separation of heavy and light nucleic acids (12), and about 0.5%–1% atom incorporation to obtain reliable mass spectroscopic detection of labeled peptides (13). As a consequence, longer incubation times and an artificially high substrate concentration can be necessary for sufficient microbial label incorporation, requiring either a larger quantity of the expensive labeled compound or an *ex situ* experimental set up where microbial communities and their activities over time may diverge from those in the original habitat. Furthermore, meta-omics approaches cannot capture the individuality of cells, thus potentially masking metabolic and spatial heterogeneity within populations (14).

These issues have been addressed, but not yet completely resolved, by coupling isotope probing with sophisticated single-cell imaging techniques, utilizing fluorescence *in situ* hybridization (FISH), or derived methods for the detection of known taxonomic and metabolic marker genes (15–19). Incorporations of the stable isotope can be visualized with nanoscale secondary ion mass spectrometry (nanoSIMS) and Raman microspectroscopy, while the incorporation of $^{14}$C has been tracked by microautoradiography (MAR) (20–25). NanoSIMS has a theoretical sensitivity of about three orders of magnitude better than protein-SIP (26, 27), but it cannot be used to gain genomic information on the organisms because the cells are destroyed in the process. In MAR, microbes that have assimilated a beta radiation-emitting substrate are detected. The radioactivity is visualized by a beta radiation-sensitive silver halide emulsion in which ionic silver is reduced to form metallic, black granules after development, revealing the distribution of radioactive isotopes (28, 29). While MAR is about three orders of magnitude more sensitive than protein-SIP and, therefore, can be carried out with concentrations in the nanomolar range that reflect *in situ* conditions (22, 30), the MAR labeling signals can, however, only be checked with a microscope and do therefore not allow for high-throughput analysis. Raman microspectrometry, on the other hand, is a non-invasive analytical tool, which can detect targeted cells with or without SIP (31–34). By combining it with single-cell sorting tools and downstream analyses, it is possible to taxonomically identify targeted cells and retrieve their genome information (35, 36). However, this method has limited screening throughput (about 10–30 s per Raman spectrum for one prokaryotic cell) and low sensitivity (about 10% atom incorporation) and can, therefore, not be applied to highly diverse communities (33, 37). A recent method advancement, stimulated Raman scattering–two-photon fluorescence *in situ* hybridization (SRS-FISH), has much higher throughput (imaging speed of 10–100 ms per cell) and sensitivity when using $D_2O$ as a label (about 2% atom incorporation) together with a carbon substrate of interest to detect metabolically active cells (34). The caveat of using $D_2O$ is that labeling is carried out in an artificial medium to ascertain that only cells of interest have detectable metabolic activity. However, incubation in an artificial medium is already a selection step and could potentially introduce bias.

Furthermore, the employment of FISH, while powerful, nevertheless has limitations. The design and selection of nucleic acid probes for FISH usually rely on educated guesses about who the target microbe might be. Due to the catabolic and genetic diversity of microbial species, it can be difficult to select representative nucleotide markers that cover all organisms performing a specific metabolism, particularly in highly diverse ecosystems like soil and sediment (38). It is also not always possible to design a specific probe that discriminates between closely related microorganisms, especially when targeting 16S rRNA as a marker. Importantly, labeling approaches employing FISH can address whether a known microbe has a function of interest but does not allow *de novo* identification. Therefore, being able to link metabolic traits to cellular identity without the requirement of prior knowledge of genes and genomes of the targeted microbial

community has become an emerging frontier research field in microbial ecology and is termed "next-generation physiology" (39).

The aim of this work was to establish a novel next-generation physiology tool that addresses the limitations of the above-mentioned methods. To this end, we have developed a workflow in which MAR is performed with cells embedded in alginate microcapsules rather than on slides. Microcapsules containing labeled cells are then identified and sorted with a fluorescence-activated cell sorter (FACS) in high throughput based on the MAR signal. A subsequent generation of single amplified genomes (SAGs) is then used to taxonomically identify the labeled cells. We name this method single cell microautoradiography and genome sequencing (scMAR-Seq) (Fig. 1). Benchmark experiments with mock communities comprising benzene degraders and non-degraders demonstrated that scMAR-Seq enables taxonomic identification of single cells selected from a community with high sensitivity based solely on their metabolic phenotype.

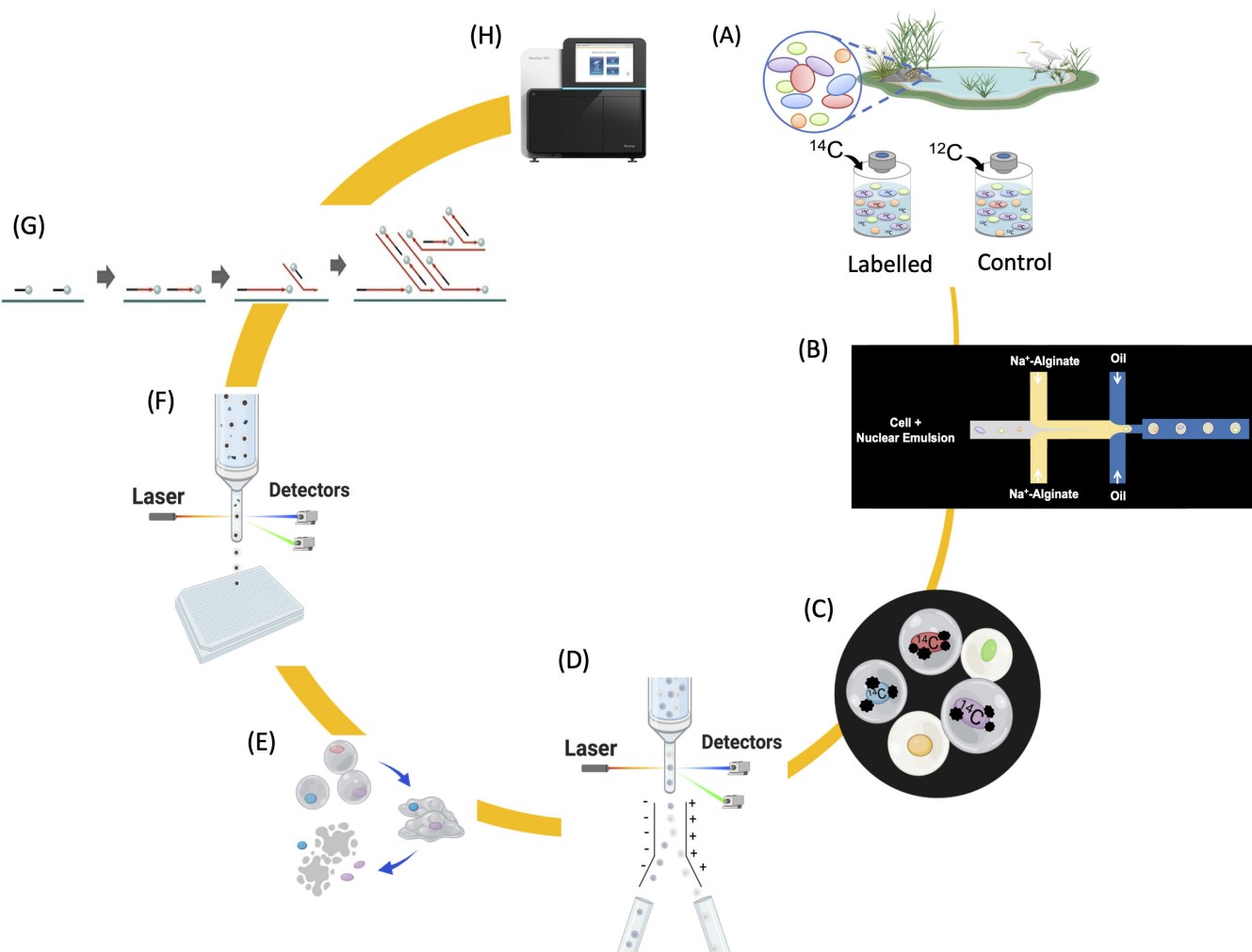

**FIG 1** Concept of scMAR-Seq. (A) Environmental samples are collected and individually incubated in parallel with radioisotope-labeled and unlabeled substrates. Steps B and C are then carried out in a dark room: (B) using a microfluidic device, individual community members are embedded in an alginate microcapsule that contains nuclear emulsion. (C) During the exposure period, radioactively labeled cells in the microcapsules generate a latent image in the nuclear emulsion, which is then developed and fixed to form black silver granules (MAR signals). (D) MAR-positive microcapsules are isolated in bulk with a cell sorter. (E) Radioisotope-labeled cells are released from the alginate microcapsules and (F) subsequently collected by a second round of cell sorting. (G) Sorted cells are lysed and their DNA is subjected to multiple displacement amplification. (H) DNA sequencing and bioinformatics analysis reveal the phylogenetic identity of cells that perform the metabolic activity of interest.

## RESULTS AND DISCUSSION

### Generation of single cell-laden microcapsules using microfluidics

The key novel feature of the method is to perform MAR in microcapsules. This allows coupling the high sensitivity of MAR with high-throughput cell sorting and the taxonomic identification power of single-cell genomics. Prior knowledge of the composition of a microbial community is not required, which is a major advance over MAR coupled with FISH (aka MAR-FISH, STARFISH, or MicroFISH [20–25, 29]). The development of the approach was modular, modifying and combining established methods to create a novel workflow and enabling the individual development and testing of the various steps in the overall procedure. Method development included several intermittent proof-of-concept trials with experimental designs not reported here for brevity, but for which documentation is available in two Ph.D. theses (40, 41).

Microfluidics was used for cell encapsulation. The key objectives were to achieve sufficient capsules stability to withstand the high pressure of subsequent high-throughput sorting (see below) and a diameter much smaller than the nozzle in the FACS (100 µm). Furthermore, since the encapsulation and the MAR process must be carried out in the dark due to the light-sensitive nuclear emulsion, it was important to devise a facile procedure that needed only a few manual steps.

First, different polymers as microcapsule matrices were evaluated. Nuclear emulsion employed in MAR usually consists of gelatine with embedded silver halide crystals. Using this emulsion as the sole matrix, we did not obtain microcapsules of sufficient stability. Next, nuclear emulsion with either agarose, polyacrylamide, or alginate was tested, all of which have been previously shown to be suitable for cell encapsulation due to their biocompatibility, porosity, and hydrophilicity (42, 43). Microcapsules with agarose were susceptible to rupture during subsequent steps of the workflow. Capsules containing polyacrylamide also proved unsuitable because of inhomogeneous distribution of the higher-density nuclear emulsion, apparently enabled by the long gelation process (20–30 min). In contrast, alginate microcapsules underwent a faster gelation transition and exhibited a more homogenous distribution of the nuclear emulsion than those with polyacrylamide and had greater stability than those with agarose. Therefore, the combination of alginate and nuclear emulsion was selected as the most suitable encapsulation matrix (for the sake of brevity, we omitted the words "nuclear emulsion" in the following when addressing capsules with alginate).

In general, two processes occur during the generation of cell-laden alginate microcapsules using a microfluidic device: (i) emulsification of a cell-alginate suspension in a continuous oil phase, and (ii) the individual gelation of alginate droplets in the oil (44). The first process is governed by the flow rate, the channel dimensions of the microfluidics device, and the chemistry of the continuous phase. These parameters were empirically determined in this study, with final values given in the Materials and Methods section.

Gelation of the alginate droplets was more difficult to control. Alginate is a copolymer of α-L-guluronic acid and β-D-mannuronic acid that forms a hydrogel via chain-crosslinking with divalent cations (45). The affinity of alginate with different divalent cations diminishes in the following order: Pb > Cu > Cd > Ba > Sr > Ca > Co, Ni, Zn > Mn (46). Among these, barium, strontium, and calcium have been demonstrated to preferentially bind to the guluronic acid to generate a so-called "egg-box" structure, forming a gel with high strength and stability (47). Here, we opted for barium and calcium. On- and off-chip gelation are the two strategies for generating alginate capsules with microfluidic devices. First, we designed and fabricated a polydimethylsiloxane (PDMS) chip for off-chip gelation in a $BaCl_2$ bath. This approach was unsuccessful due to oil droplet coalescence occurring in the collection bath, despite the use of a strong surfactant. Thus, most of the alginate microcapsules obtained were inhomogeneous in size and shape and carried more than one cell. The main challenge in performing on-chip gelation is that the kinetics of ionic cross-linking is governed by the intrinsic and rapid binding

of the gelling ions by the ionotropic polymer (48). Premature gelation caused by a rapid ion-alginate crosslinking reaction can block microchannels and outlet channels in the microfluidic chip or produce nonuniform droplets (48–50). In order to prevent clogging and capsule coalescence due to premature alginate solution-to-gel transition, we designed a chip in which there was a lag between the emulsification and gelation (Fig. S1). This was accomplished by separating droplet formation and internal gelation in time and space while keeping all encapsulations still occurring in the flow stream on the chip. The lag was created by prolonging the diffusion of the crosslinking ions, thus making them less accessible to alginate chains until the droplet formation was completed. To obtain the temporal separation, we adapted the competitive ligand exchange crosslinking technique (CLEX) (51). By applying CLEX, the diffusion rate of the inorganic cations as gelling ions is controlled by the competition between the ionotropic polymer and anionic chelates with different equilibrium binding constants. While the original CLEX approach uses zinc cation ethylenediaminediacetic acid (EDDA) alginate solution and calcium cation EDTA alginate solution, we substituted $Ca^{2+}$ with barium cation as the gelling ion since the barium-alginate microcapsules better retained their spherical morphology during the subsequent autoradiography and sorting procedures. The results of calcium and barium CLEX gelation kinetics are listed in Table S1. Furthermore, a

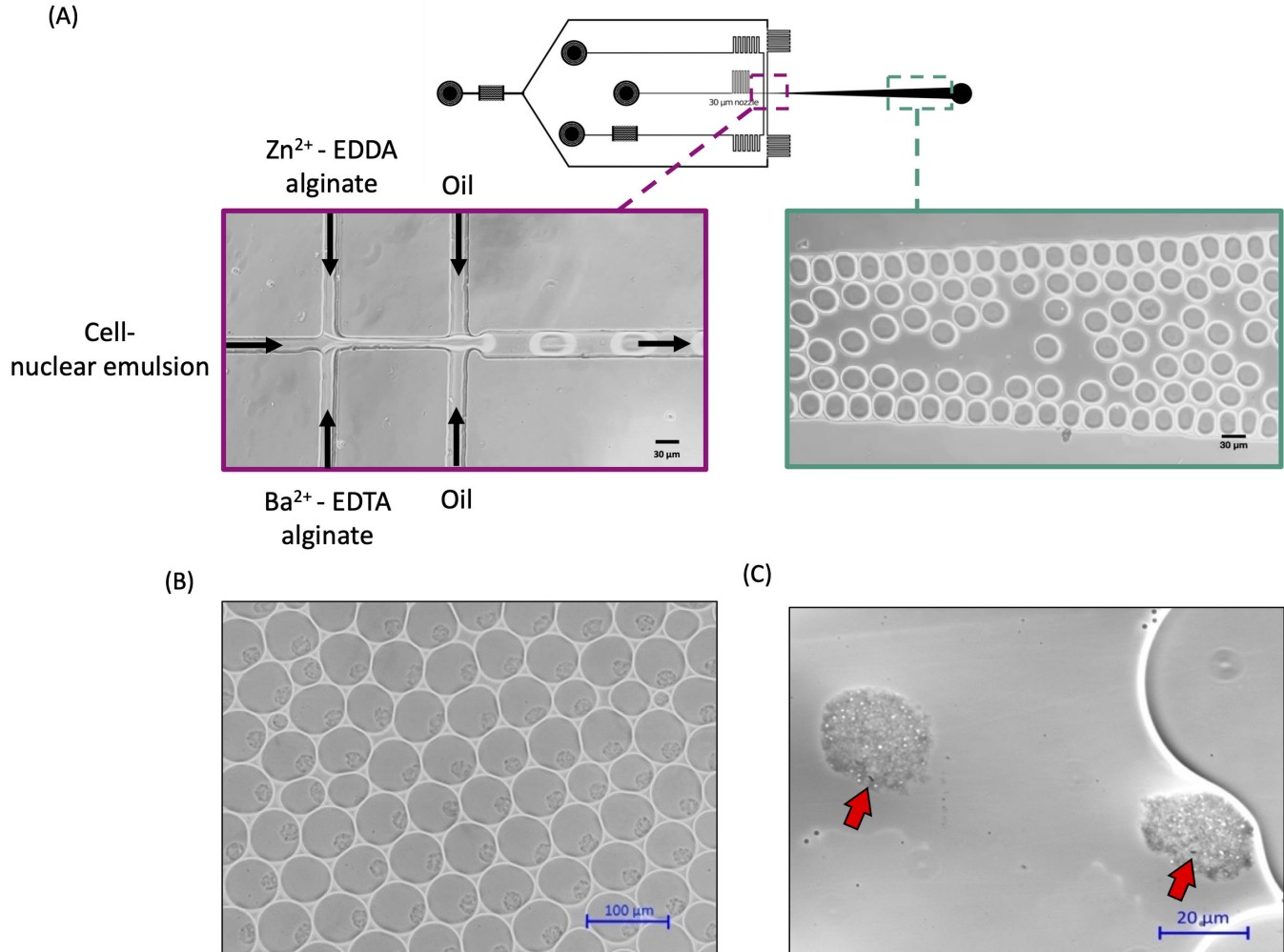

FIG 2 Fabrication of single-cell laden MAR alginate microcapsules. (A) Schematic image of the microfluidic chip, the fluidic scheme (violet frame), and droplet formation on the chip (green frame). The cell-nuclear emulsion mixture is introduced through the central channel and meets the two alginate precursor solutions at the first cross junction. Alginate-in-oil droplets are then formed upon encounter with the carrier oil and are transported to the collection outlet. (B) Micrograph of the collected alginate-in-oil droplets. (C) Each alginate microcapsule is 20–25 µm in diameter and contains a single bacterial cell (red arrow).

co-flow scheme was designed to create spatial separation between the two alginate precursors until the alginate-in-oil droplet formation. As illustrated in Fig. 2A; Movie S1, the nuclear emulsion containing the cells enters the chip through the middle microchannel, which forms a first cross-junction with two channels containing the individual CLEX solutions, gelling ion solution (i.e., $Ba^{2+}$-EDTA-alginate), and exchange ion solution (i.e., $Zn^{2+}$-EDDA-alginate). The middle aqueous stream acts as a blocking stream to prevent premature gelation of the two CLEX solutions at the first cross-junction. When the co-flow reaches the second cross-junction where it encounters the carrier oil to form alginate-in-oil droplets, the chelating competition and ion exchanges trigger the sol-gel transition within each oil droplet while rolling along the microchannel until being harvested at the outlet port on the chip (Movie S2).

With this chip design and encapsulation strategy, the microfluidic system was able to generate stable single-cell laden alginate microcapsules with diameters around 20–25 µm (Fig. 2B), which we deemed a suitable size for subsequent sorting with our FACS. We adjusted the initial input bacterial cell density between $10^5$ and $10^6$ cells mL$^{-1}$ to ensure each alginate microcapsule contains at most only a single microbial cell (Fig. 2C). The encapsulation procedures could be continuously operated for at least 5 hours in the darkroom without clogging issues.

## Microautoradiography in alginate microcapsules

To establish the protocol for carrying out MAR in the microcapsules, [$^{14}$C] benzene was selected as the radioactive substrate, and *Pseudomonas veronii* strain B560 as the benzene-degrading bacterium. As discussed previously in detail (29), the substrate concentration and incubation time in MAR-labeling are chosen according to the specific metabolic activity of the investigated microbial community, ranging from low nanomolar to low millimolar and typically lasting a few hours. Here, a 1-hour pre-incubation of *P. veronii* with 0.12 mM $^{12}C_6H_6$ prior to the $^{14}C_6H_6$ incubation was carried out in order to ensure that degradation occurred. The decrease in benzene concentration was checked by gas chromatography. This pre-incubation was necessary because a lag often occurred after the addition of benzene to the cultures of *P. veronii* B560. The generation time of this strain with 0.24 mM benzene also varied between 2.5 and 10 hours. After confirming that benzene degradation had occurred during pre-incubation, an additional 0.12 mM (750 kBq equals 20.3 µCi) of radioactive [$^{14}$C] benzene was added together with an equal concentration of non-radioactive [$^{12}$C] benzene and incubated for 2 hours. The MAR-negative control was prepared by using *Escherichia coli* K12, which cannot degrade benzene. Since benzene is moderately lipophilic, a triple wash step was performed after both the *E. coli* and *P. veronii* incubations to remove residues that had accumulated in the membrane lipid bilayer. Measuring the radioactivity via liquid scintillation counting confirmed that none of the *E. coli* in the negative control retained [$^{14}$C] benzene. The blank control was prepared by incubating *P. veronii* with regular $^{12}C_6H_6$.

Cells of the three different treatments were encapsulated individually as positive, negative, and blank control using the PDMS chip. The generated and collected capsules were sealed in a dark box for MAR exposure. Exposure lengths of 3–7 days were applied according to previously published time frames (29). During exposure, the capsules were still within the oil droplets.

Common developers in MAR use reducing agents such as hydroquinone, metol, amidol, and *p*-phenylendiamine at alkaline pH together with additives for process optimization and preservation. All of these developers caused substantial swelling or destruction of the capsules. Therefore, we adapted Kodak's "Professional Xtol developer," which uses isoascorbate and dimezone S (4-hydroxymethyl-4-methyl-1-phenyl-3-pyrazolidinone) at circumneutral pH as reducing agents. Ingredients in Xtol that increase the shelf life of the developer solution had a negative effect on the stability of the capsules and were therefore not included. Furthermore, $BaCl_2$ was added to the developer to stabilize the alginate capsules. For the fixer, a modified recipe with a comparatively low concentration of sodium thiosulfate concentration was used, as the high

concentration in the original fixer recipe compromised capsule stability, probably due to the removal of barium from alginate by replacement with sodium as well as complexation with thiosulfate. Furthermore, the fixer was amended with a mixture of $BaCl_2$ and $CaCl_2$ for increased capsule stability. Using $Ba^{2+}$ alone in the respective high concentration sometimes led to the formation of large crystals inside capsules, presumably poorly soluble Ba-thiosulfate, resulting in their mechanical destruction.

The modified developer and fixer were tested using cell-loaded capsules post-exposure and removal of the oil with perfluorooctanol, followed by microscopic inspection (Fig. 3). As in MAR-FISH (29), we differentiated between MAR-positive and MAR-negative signals by a qualitative assessment of the number of black silver granules per capsule. Quantitative MAR is possible on microscopy slides with thinly coated nuclear emulsion (22) but was not carried out in this study since the elucidation of the sensitivity threshold of the approach was not an aim at this stage of method development. Furthermore, determining the number of silver granules would have been error-prone given the diameter of the capsules, and elucidating the qualitative correlation of the number of granules with the FACS signal used to sort the capsules (see next section) would have been rather difficult. Figure 3 shows that microcapsules with $^{14}$C-labeled *P. veronii* had a much higher abundance of silver granules than the background in capsules from the negative and blank controls. The signal intensity increased with the exposure duration in the positive microcapsules, while the background signal in the capsules with the negative and blank controls remained approximately the same throughout the 7 days of exposure.

As in MAR-FISH (29), the optimal exposure time in scMAR-Seq needs to be empirically determined for each sample and substrate by regular inspections of silver granule density per capsule. While insufficient exposure time would hamper the detection of sufficient numbers of active cells during subsequent sorting, a too-long exposure

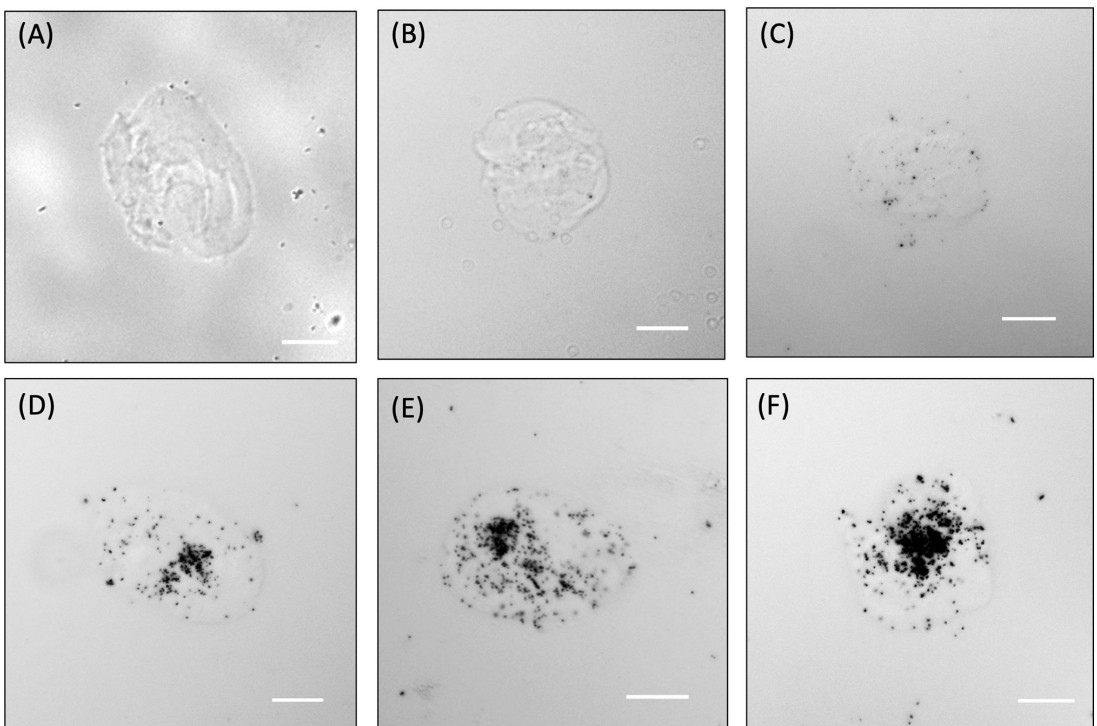

**FIG 3** Comparison of the MAR images of alginate microcapsules generated from the blank (*P. veronii* cells without [$^{14}$C] benzene), negative (*E. coli* cells with [$^{14}$C] benzene), and positive control (*P. veronii* cells with [$^{14}$C] benzene). Individual fractions of the three treatments were taken after different exposure durations, developed, fixed, and viewed under a microscope. There was no MAR signal (black granule cluster) in the (A) MAR-blank control, and the negative control with (B) 3 and (C) 7 days of exposure. For the MAR-positive control, the black granule cluster increased gradually from (D) 3 days to (E) 5 days to (F) 7 days of exposure. Scale bars: 10 μm.

time could increase the background signal and thus might yield false-positive results, especially at low *in situ* turnover rates and low cell numbers in the sample.

## Sorting and multiple displacement amplification

Next, a FACS protocol to sort microcapsules based on their optical properties was established. The advantage of a FACS for application in scMAR-Seq is its ability for automated, high-throughput, and multi-parameter-characterization sorting (52). Voltage settings of the photomultiplier tube (PMT) of the FACS were optimized to ensure that most of the alginate capsules were present within the bounds of the forward scatter (FSC)-height vs side scatter (SSC)-height scatterplots. By using an SSC-height vs SSC-area gating strategy, microcapsules with a high density of silver granules could be clearly gated and distinctly differentiated from the capsule population without MAR signal (Fig. 4; Fig. 5A and B).

To test the specificity of the sorting and to conduct single-cell genomics, a previously modified and improved multiple displacement amplification (MDA) method called WGA-X (53) was applied to the sorted microcapsules. At first, MDA was unsuccessful (0%, $n = 76$) with both positive and negative capsules, i.e., regardless of the GC content of the bacterial genome or the presence of silver granules. The lack of DNA amplification from encapsulated cells was likely due to various factors such as the alginate capsule preventing efficient cell lysis and hence reducing the amplification effectiveness as well as inhibition of the DNA polymerase by various MAR components. To overcome this problem, we adapted a dual-sorting strategy (54). First, positive MAR microcapsules were sorted in bulk. The alginate structure was then removed by chelating the $Ba^{2+}$ from the gel matrix using EDTA to release the encapsulated radiolabeled cells, which were then collected individually in microtiter plates in a second sort. For this sorting, the PMT voltage of the FACS was adjusted to the instrument's settings for bacterial cells with *P. veronii* and *E. coli* as standards. Sorting was based on the signal intensity of the FSC area vs the SSC area (Fig. 4). The sorting gate was set such that the bacterial cells were well separated from the alginate debris of the lysed microcapsules, which constituted the majority of particles. With the dual sort approach, the MDA success rate ranged from

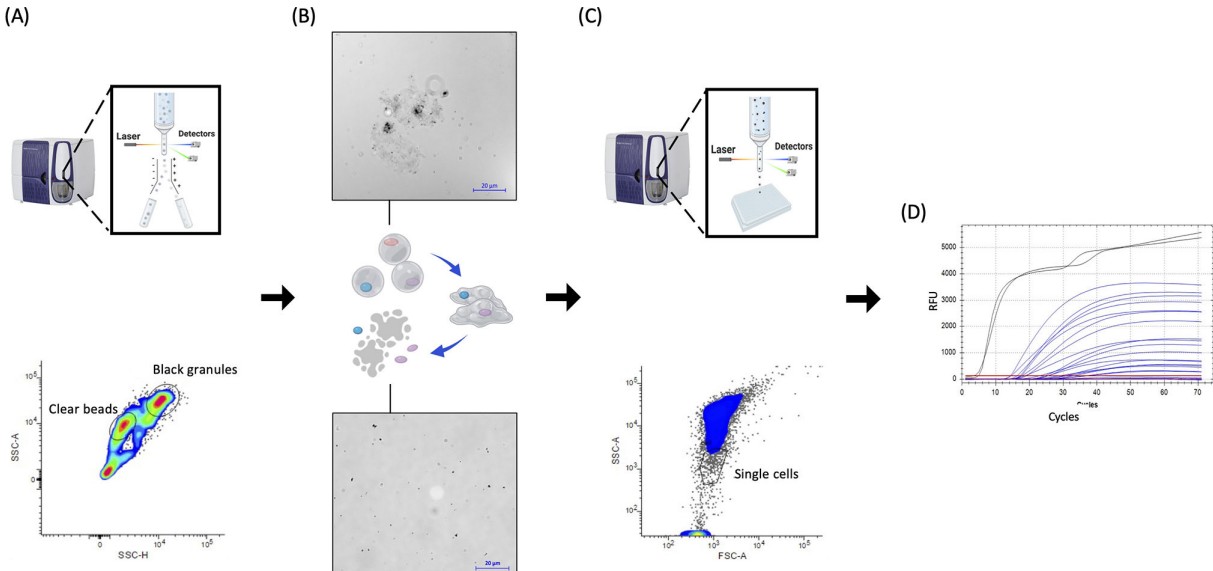

**FIG 4** The dual-sorting operation procedures of scMAR-Seq. (A) The MAR microcapsules with black granules were first sorted in bulk. (B) The sorted microcapsules were subsequently lysed by using EDTA, and then the fragmented alginate particles could be clearly observed under the microscope. (C) A second FACS sorting was performed to sort the liberated targeted single bacterial cells, followed by (D) multiple displacement amplification using WGA-X. The black curves in the WGA-X result graphs represented the amplification from the DNA positive control and the blue curves were the amplification from single bacteria cells. Throughout the entire reaction, no amplification from the negative control (red curves) was detected.

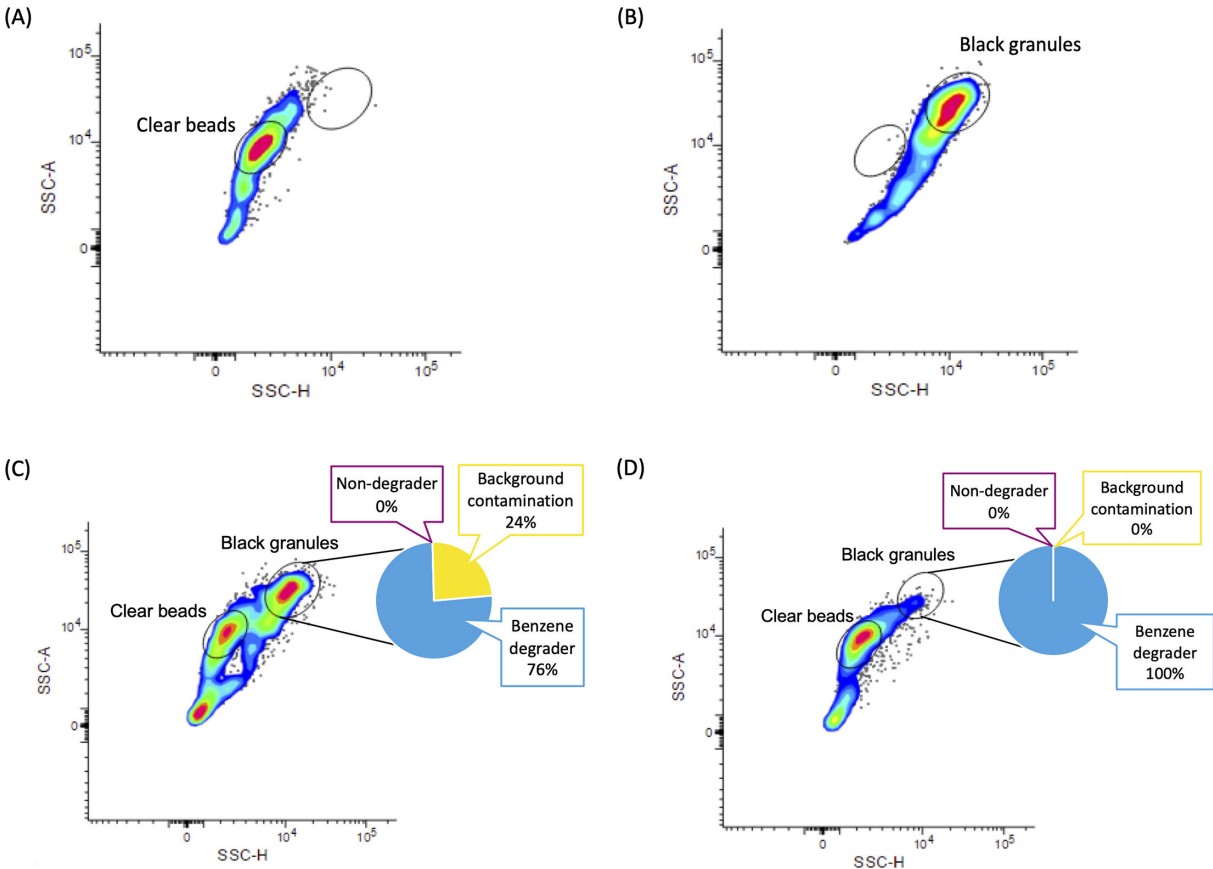

**FIG 5** Representative graphs for FACS analysis of the MAR microcapsules. In the side scatter-area (SSC-A) and side scatter-height (SSC-H) detection mode, the gates labeled "Clear beads" and "Black granules" define the population of MAR-negative and MAR-positive microcapsules, respectively. "Background contamination" refers to capsules where all amplified DNA was derived from typical multiple displacement amplification contaminations, alginate, or gelatine (see Materials and Methods). FACS scatter graphs depicting the distribution of MAR-positive and MAR-negative populations in (A) the negative control, *E. coli* cells with [$^{14}$C] treatment; (B) the positive control, *P. veronii* cells with [$^{14}$C] treatment; the proof-of-principle experiment with the (C) 50%–50% mock community, and (D) 10%–90% mock community.

10% to 50% (*n* = 185) according to the now possible generation of *P. veronii* SAGs after library preparation and sequencing (Table S2, and SAGs plus detailed assembly statistics on Figshare [10.6084/m9.figshare.24131106]).

## Applying scMAR-Seq on two mock communities

The scMAR-Seq pipeline was tested with two mock communities: one contained 50% degrader (*P. veronii*) and 50% non-degrader (*E. coli*), and the other contained 10% degrader and 90% non-degrader. As shown in the FACS scatter plots, the results of the first sort showed distinct populations for MAR microcapsules containing radioisotope-labeled cells and non-labeled cells on both 50%–50% (Fig. 5C) and 10%–90% (Fig. 5D) mock communities.

Subsequently, MDA was performed on the lysed single cells that were collected after the second sort. The accuracy of the targeted sorting was evaluated by sequencing SAG libraries and assembling individual genomes to determine whether *P. veronii* or *E. coli* were sorted and amplified. The evaluation was done by mapping reads to the reference genomes and an MDA contamination database constructed in our research group (see Materials and Methods) and, in addition, determined by the best BLAST-hit of the *de novo* assemblies. The background contamination that was present in SAGs from positive MAR microcapsules was less than 0.4% of total reads on average, a neglectable value for SAGs (55), either from intrinsic material-derived contaminants, which are specific to MAR

microcapsules, or typical DNA contamination that is frequently found in laboratories and reagents.

After filtering out background contaminant reads, individual SAGs were assembled *de novo*. The genome completeness and taxonomy identification were obtained by using MDMcleaner (56) and an additional BLASTn search against NCBI's non-redundant nucleotide database after excluding short contigs <5 kb in length. Among the 58 SAG libraries from the 50%–50% community, 44 were revealed as *P. veronii*, while 14 SAGs contained non-identifiable sequences that were probably generated during library preparation or MDA-derived artefacts, but no *E. coli*. Six SAGs were obtained from the 10%–90% community, all of which were derived from *P. veronii*. Maximum genome completeness was 11.4% (Table S2, and SAGs plus detailed assembly statistics on Figshare [10.6084/m9.figshare.24131106]). It is conceivable that the low genome completeness was at least partially due to DNA damage occurring during scMAR-Seq but should be sufficient for the taxonomic identification of most bacterial cells, as was shown here with *P. veronii*.

## Conclusion and outlook

Elucidating which microbial community members perform a particular metabolic function is a central challenge in microbiology. A suite of methods can provide insights into community structures; however, all of them have some analytical limitations in linking taxonomy to *in situ* function, have low sensitivity, or are not suitable for high-throughput analyses (19, 39). The here-introduced scMAR-Seq workflow combines the sensitivity of MAR with the miniaturization of microfluidics, the high-throughput sorting capability of a FACS, and taxonomic resolution of single-cell genomics.

The pipeline was established with $^{14}C$-labeled benzene as a test substrate, but as in classical MAR-FISH (29), molecules with other radioisotopes emitting weak beta-radiation such as $^{3}H$ (tritium), $^{35}S$, and $^{33}P$ could be used as well. When employing the pipeline with complex environmental samples and other organic substrates in the future, knowledge of the *in situ* turnover rate of the selected compound could help to limit the labeling time needed. The incubation time during MAR depends on the degree of assimilation of the labeled substrate and thus will have to be determined empirically in future experiments by tracking the incorporation of radioactivity into total microbial biomass. The same guidelines for concentrations and labeling time that were formulated for MAR-FISH apply to scMAR-Seq (29).

While we envision that scMAR-Seq can yield novel insights into microbial ecology, it should also be noted that it is a workflow that requires access to a dark room and a certified radioisotope laboratory. Future improvements of scMAR-Seq will target MDA efficiency and genome recovery, which are common issues in single-cell genomics in general (7). Nevertheless, unambiguous taxonomic identification could be achieved even in cases of low genome completeness. Therefore, this pipeline is not only capable of revealing the metabolic activity of known species but also has great potential to investigate and explore novel functional features and genes from microbial dark matter, aiding in the use of microorganisms for future biotechnological applications.

## MATERIALS AND METHODS

### Fabrication of PDMS microfluidic chips

The microfluidic chips used throughout this research were designed and manufactured in-house. Figure S1 provides a visual depiction of the newly developed chip design. The CAD file of the chip is available online (10.6084/m9.figshare.23735268). After printing an inverted image on a transparent foil (DTP-system Studio), the master wafer was produced using soft lithographic techniques. The first step was to apply a 30 µm thick layer of the negative photoresist SU-8-2050 (MicroChem) on a clean 4-inch silicon wafer (MicroChemicals). The coated wafer was then soft baked at 65°C for 1 min and at 95°C for

7 min before proceeding with UV exposure. The post-exposure bake took place directly after UV exposure at 65°C for 1 min and at 95°C for 5 min. Immediately after UV exposure, the silicon wafer was immersed in the microfluidic developer (mr-Dev600, MicroChem) and gently shaken for 5 min to develop the chip pattern. After wiping with isopropanol, the SU-8 mold was hard-baked at 150°C for another 10 min. The PDMS prepolymer and curing agent (Sylgard 184 silicone elastomer kit, Dow Corning) were mixed in a 10:1 ratio, poured over the SU-8 mold to form the upper layer, and poured over another empty, clean silicon wafer to form the bottom layer. After curing at 100°C for 3 hours, the cured PDMS replica was peeled off and the inlets and outlets were punched in using a 1 mm biopsy puncher. Then, the PDMS replica and a complementary PDMS layer were exposed to oxygen plasma (100 W, 500 mTorr, 30 s, Diener electronic) and were bonded immediately (57).

## Isotope labeling of bacterial cultures

All solutions were prepared using ultrapure deionized water (DI) with a resistance of 15 MΩ cm (Milli-QTM, Millipore). Pure cultures of *P. veronii* B560 (strain collection of the UFZ, Leipzig) and *E. coli* K12 were grown in liquid mineral medium M9 with trace elements (0.1%, vol/vol) and vitamins (58). Cells used for isotope labeling were harvested from fresh cultures with 0.24 mM [$^{12}$C] benzene (HPLC grade, ≥99.9%) dissolved in 2,2,4,4,6,8,8-heptamethylnonane for *P. veronii* and with 0.24 mM of glucose for *E. coli* as respective growth substrate. Since benzene is a volatile compound, the cells were cultured in glass vials sealed with air-tight rubber stoppers. After harvesting, cells were resuspended in 2 mL of fresh M9 medium in a 10 mL serum bottle that contained 0.12 mM of [$^{12}$C] benzene for both *P. veronii* and *E. coli* to a final cell density of ca. $2 \times 10^8$ cells mL$^{-1}$. Incubation was carried out at 30°C on a rotary table at 150 rpm for 1 hour prior to the addition of the radioisotope. Isotope labeling of the cells was conducted by incubating the harvested cells with 0.12 mM of [$^{14}$C] benzene (3,552 MBq/mmol, American Radiolabeled Chemicals) and 0.12 mM of [$^{12}$C] benzene at 30°C on a rotary table set at 150 rpm for 2 hours with oxygen as an electron acceptor provided by the airspace, and the initial cell number was adjusted to $10^8$ cells mL$^{-1}$ in the 2 mL total volume. Following isotope labeling, cells were harvested by centrifugation at $17,000 \times g$ for 10 min and three washes with 40 mM MOPS (3-morpholinopropane-1-sulfonic acid) buffer (pH 7.4) before being stored in MOPS buffer at 4°C until encapsulation.

## On-chip gelation of single-cell laden alginate microcapsules

To generate MAR microcapsules, two alginate precursor solutions were prepared. The gelling ion precursor solution was prepared by mixing alginate and Ba$^{2+}$-EDTA solution (0.125 M BaCl$_2$ and 0.125 M EDTA, pH 6.7) to obtain a final concentration of 0.25% (wt/vol) alginate, 94 mM of Ba$^{2+}$, and 94 mM of EDTA. The exchange ion precursor solution was prepared by mixing alginate with Zn$^{2+}$-EDDA solution [0.5 M Zn(CH$_3$COO)$_2$ and 0.5 M EDDA, pH 6.7] to a final concentration of 0.8% (wt/vol) alginate, 300 mM Zn$^{2+}$, and 300 mM EDDA. All the solutions mentioned above as well as the 40 mM MOPS buffer required decontamination by autoclaving, followed by UV irradiation for 6 hours. To prepare the light-sensitive nuclear emulsion, the following operation was performed in a dark room with dark room safe light (Kaiser) covered with an ILFORD 902 filter (ILFORD). First, the K5 nuclear emulsion (ILFORD) was melted in a 42°C water bath. The melted K5 emulsion was then mixed with 40 mM MOPS in a 1:1 volume ratio. The [$^{14}$C] benzene-exposed cells were then mixed with the nuclear emulsion to a final cell density of ca. $1 \times 10^6$ cell mL$^{-1}$. The carrier phase in microfluidic encapsulation was fluorinated oil HFE7500 containing 2% (vol/vol) of the surfactant Pico-Surf1 (Dolomite). Flow rates were determined based on empirical testing: the two alginate precursor solutions, the K5-cell mixture, and the carrier phase were set at 1, 1.2, and 2.3 µL h$^{-1}$, respectively. The MAR microcapsules were stored in a dark box and exposure was carried out at room temperature for 3–7 days.

## Microautoradiography

The microcapsules were autoradiographically processed under darkroom conditions (Fig. S2). To maintain the rigidity and integrity of the alginate microcapsules, all solutions used in classical MAR with high pH were replaced with circum-neutral alternatives. First, a 10 µm CellTrics cell strainer (Sysmex) was placed on a clean 1.5 mL microcentrifuge tube. The alginate microcapsules and the oil layer were then transferred to the cell strainer, and 180 µL of 1H,1H,2H,2H-perfluorooctanol (Sigma-Aldrich) was evenly distributed over the filter to break the oil droplets, which took approximately 1 min. Next, the filter was carefully inverted and placed on a 24-well plate to backwash the alginate capsules from the filter into the plate with 400 µL DI water, followed by transferring the suspension with the capsules to a clean 1.5 mL microcentrifuge tube. The suspension was then centrifuged at 900 $g$ for 3 min to sediment the alginate capsules. After discarding the upper aqueous layer, the microcapsules were washed again with DI water and centrifuged to remove the upper aqueous layer.

For MAR development, a modified developer was made based on Kodak's commercial "Professional Xtol developer" containing 6.5 g sodium isoascorbate, 0.2 g 4-(hydroxy-methyl)–4-methyl-1-phenyl-3-pyrazolidinone, 1.2 g Tris-base, and 12.21 g $BaCl_2$ per liter. A total of 400 µL of the developer was added to the MAR microcapsules and mixed thoroughly by pipetting gently up and down. The microcapsules were developed for 3 min before centrifugation at 900 $\times g$ for 3 min. After discarding the supernatant, the capsules were washed twice with 400 µL of DI water. Subsequently, 400 µL of fixer containing 170.76 mM sodium thiosulfate, 13.64 mM $CaCl_2$, and 6.8 mM $BaCl_2$ in DI water was added to the capsules and again mixed thoroughly by pipetting gently up and down. The capsules were then left to react for 3 min and centrifuged at 900 $\times g$ for 3 min. After discarding the supernatant, the capsules were washed twice with 400 µL DI water and centrifuged at 900 $\times g$ for 3 min, re-suspended, and stored in 250 mM $BaCl_2$ at room temperature until sorting.

## Cell sorting

All capsules and cells were sorted using a BD FACSMelody Cell Sorter (BD Biosciences) with a 100 µm sorting nozzle. The sheath fluidic was 1× PBS (phosphate-buffered saline) made from 5× PBS stock in DI water, which was autoclaved and then UV irradiated for 6 hours. The alginate microcapsules that exhibited a positive MAR-signal were first sorted into 5 mL Falcon polypropylene tubes (Corning) using purity sort mode. PMT voltage settings were optimized to ensure that most capsules (diameter around 20–25 µm) were present within the bounds of the scatterplot of forward scatter-height vs side scatter-height. For the first sort, alginate capsules were gated based on SSC-height and SSC-area values. The sorted capsules were then lysed with 150 mM EDTA and vortexed vigorously for 5 min to release radioactive-labeled cells from the alginate. For the second sort, the liberated cells were sorted into Hard-Shell 384-well plates (Bio-Rad Laboratories) using single-cell mode. PMT voltage was adjusted to the settings for bacterial cells, and the cells were sorted based on the signal intensity of FSC-area vs SSC-area.

## Single-cell genome amplification by WGA-X

Prior to single-cell genome amplification, cells were lysed by one freeze-thaw cycle at −80°C and then at 20°C. The amplification procedures followed the WGA-X protocol (53), except that the total volume was 5 µL instead of 10 µL. In brief, the WGA-X components are as follows: 0.2 U µL$^{-1}$ Equiphi29 polymerase (Thermo Fisher Scientific), 1× Equiphi29 reaction buffer (Thermo Fisher Scientific), 10 mM dithiothreitol (Thermo Fisher Scientific), 40 mM Exo-Resistant Random Primer (Thermo Fisher Scientific), 0.4 mM dNTP (New England BioLabs), and 1 µM SYTO-13 (Thermo Fisher Scientific). The WGA-X reaction was carried out by using a CFX384 TouchTM Real-Time Detection System (Bio-Rad Laboratories) for 7 hours at 45°C and then inactivated by incubation at 75°C for 15 min.

## Library preparation, single-cell genome sequencing, and bioinformatics analyses

Purification of the WGA-X products was done by using the DNA Clean & Concentrator-5 according to the manufacturer's manual (Zymo Research). SAG libraries were prepared for paired-end sequencing on an Illumina NexSeq 550 (2 × 150 bp) using the NEBNext UltraTM II FS DNA Library Prep Kit (New England BioLabs) following the manufacturer's instructions. Library qualities were evaluated using the Agilent High Sensitivity DNA Kit on the Agilent 2100 Bioanalyzer instrument (Agilent Technologies). Low-quality reads and adaptor sequences were removed by using a three-step process, consisting of Trimmomatic v.0.36 (59), bbduk v.35.69 (60), and cutadapt v.1.14 (61). The overlapping read pairs were then merged using FLASH v.1.2.11 (62). The detailed arguments used for read trimming and merging were previously described (54). The trimmed sequences from each SAG were aligned against the reference genomes of *P. veronii* B560 (GenBank accession number: JAODYJ000000000) and *E. coli* K-12 MG1655 (GCF_000005845.2) by using BWA-MEM v 0.7.17 (63).

The unmapped reads were then extracted by using SAMtools v 1.9 (63) to identify potential background contaminants. The genomes of *Bos taurus* (GCF_002263795.1), *Sus scrofa* (GCF_000003025.6), *Ectocarpus siliculosus* (CABU00000000.1), *Macrocystis pyrifera* (JAALFD000000000.1), *Paenibacillus polymyxa* (GCF_006274405.1), *Meiothermus ruber* (GCF_000376665.1), *Stenotrophomonas maltophilia* (GCF_900475405.1), *Rhizobium rhizogenes* (GCF_018138105.1), *Achromobacter xylosoxidans* (GCF_008432465.1), and *Homo sapiens* (GCF_000001405.40) were downloaded from the NCBI database for constructing a contamination database. Background contamination was evaluated by using FastQ Screen (64). The reads derived from background contamination in each SAG were subsequently removed by SAMtools. The remaining reads were assembled *de novo* using SPAdes (v3.10.1) (65) with the --sc and --careful flag, and *k*-mers iteration from 21 to 101 in a step size of 10. Contigs < 5 kb were excluded from subsequent analyses. The genome recovery, taxonomy identification, and the contamination-free assemblies were obtained by using MDMcleaner (56), with an additional BLASTn examination of the closest matching genome against the NCBI's nucleotide database (nr/nt) (66).

## ACKNOWLEDGMENTS

The authors are grateful to Dr. Uwe Kappelmeyer from the Department of Environmental Biotechnology at the UFZ for assisting with the radioisotope experiment, David Thiele for laboratory assistance, Dr. John Vollmers and Dr. Florian Lenk for their excellent support on the bioinformatics, and Dr. Morgan Sobol (all from the IBG-5, KIT) for FACS technique consultation and helpful discussions.

H.-Y.L. was supported by a doctoral scholarship from the German Academic Exchange Service (DAAD). Funding for the open access charge was from institute funds.

## AUTHOR AFFILIATIONS

[1]Institute for Biological Interfaces (IBG-5), Karlsruhe Institute of Technology, Eggenstein-Leopoldshafen, Germany
[2]Department of Environmental Biotechnology, Helmholtz Centre for Environmental Research-UFZ, Leipzig, Germany
[3]Institute for Analytical Chemistry, Leipzig University, Leipzig, Germany

## AUTHOR ORCIDs

Jochen A. Müller http://orcid.org/0000-0002-0935-1523
Anne-Kristin Kaster http://orcid.org/0000-0002-6361-9551

## AUTHOR CONTRIBUTIONS

Hao-Yu Lo, Formal analysis, Methodology, Writing – original draft | Konstantin Wink, Resources | Henrike Nitz, Data curation, Formal analysis, Methodology | Matthias Kästner, Resources | Detlev Belder, Resources | Jochen A. Müller, Supervision, Writing – review and editing.

## DATA AVAILABILITY

The CAD file of the PDMS chip is available at Figshare under the DOI 10.6084/m9.figshare.23735268.The draft genome assembly of *P. veronii* B560 is available at GenBank under the accession number GCA_025421655.1. The SAGs can be found in figshare under 10.6084 /m9.figshare.24131106.

## ADDITIONAL FILES

The following material is available online.

### Supplemental Material

**Supplemental material (mSystems00998-23-S0001.docx).** Supplemental tables, figures, and movie legends.
**Video S1 (mSystems00998-23-S0002.mp4).** PDMS microfluidic chip in operation for scMAR-Seq.
**Video S2 (mSystems00998-23-S0003.mp4).** Oil droplets rolling along the microchannel towards the outlet port of the PDMS chip.

### Open Peer Review

**PEER REVIEW HISTORY (review-history.pdf).** An accounting of the reviewer comments and feedback.

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
