## [Reviewer comments · mSystems]

scMAR-Seq: A novel workflow for targeted single-cell genomics of microorganisms using radioactive labelling

Hao-Yu Lo, Wink Konstantin, Henrike Nitz, Matthias Kästner, Detlev Belder, Jochen Müller, and Anne-Kristin Kaster

Corresponding Author(s): Anne-Kristin Kaster, Karlsruher Institut für Technologie

Review Timeline:

Submission Date:

September 20, 2023

Accepted:

October 9, 2023

Editor: Matthias Hess

Reviewer(s): The reviewers have opted to remain anonymous.

Transaction Report:

DOI: <https://doi.org/10.1128/mSystems.00998-23>

October 9, 2023

Dr. Anne-Kristin Kaster
Karlsruher Institut für Technologie
Eggenstein-Leopoldshafen
Germany

Re: mSystems00998-23 (scMAR-Seq: A novel workflow for targeted single-cell genomics of microorganisms using radioactive labelling)

Dear Dr. Anne-Kristin Kaster:

Congratulations. Your manuscript has been accepted, and I am forwarding it to the ASM Journals Department for publication. However, please make sure to make the few minor edits suggested by Reviewer #2 before the manuscripts goes into production

Your manuscript has been accepted, and I am forwarding it to the ASM Journals Department for publication. For your reference, ASM Journals' address is given below. Before it can be scheduled for publication, your manuscript will be checked by the mSystems production staff to make sure that all elements meet the technical requirements for publication. They will contact you if anything needs to be revised before copyediting and production can begin. Otherwise, you will be notified when your proofs are ready to be viewed.

If you would like to submit a potential Featured Image, please email a file and a short legend to msystems@asmusa.org. Please note that we can only consider images that (i) the authors created or own and (ii) have not been previously published. By submitting, you agree that the image can be used under the same terms as the published article. File requirements: square dimensions (4" x 4"), 300 dpi resolution, RGB colorspace, TIF file format.

We recognize that the video files can become quite large, and so to avoid quality loss ASM suggests sending the video file via <https://www.wetransfer.com/>. When you have a final version of the video and the still ready to share, please send it to mSystems staff at msystems@asmusa.org.

Sincerely,

Matthias Hess
Editor, mSystems

Journals Department
E-mail: mSystems@asmusa.org